# Cellulose Nanocrystal Reinforced Chitosan Based UV Barrier Composite Films for Sustainable Packaging

**DOI:** 10.3390/polym12010202

**Published:** 2020-01-13

**Authors:** Mithilesh Yadav, Kartik Behera, Yen-Hsiang Chang, Fang-Chyou Chiu

**Affiliations:** 1Department of Chemical and Materials Engineering, Chang Gung University, Taoyuan 333, Taiwan; b.kartik1991@gmail.com; 2Department of Chemistry, Prof. Rajendra Singh Institute of Physical Sciences for Study and Research, V.B.S Purvanchal University Jaunpur, Siddikpur U.P. 222002, India; 3Department of General Dentistry, Chang Gung Memorial Hospital, Taoyuan 333, Taiwan; cyh4714@hotmail.com; 4Graduate Institute of Dental and Craniofacial Science, Chang Gung University, Taoyuan 333, Taiwan

**Keywords:** nanocomposite, chitosan, cellulose nanocrystal, UV barrier, mechanical properties

## Abstract

In this study, green composite films based on cellulose nanocrystal/chitosan (CNC/CS) were fabricated by solution casting. FTIR, XRD, SEM, and TEM characterizations were conducted to determine the structure and morphology of the prepared films. The addition of only 4 wt.% CNC in the CS film improved the tensile strength and Young’s modulus by up to 39% and 78%, respectively. Depending on CNC content, the moisture absorption decreased by 34.1–24.2% and the water solubility decreased by 35.7–26.5% for the composite films compared with neat CS film. The water vapor permeation decreased from 3.83 × 10^−11^ to 2.41 × 10^−11^ gm^−1^ s^−1^Pa^−1^ in the CS-based films loaded with (0–8 wt.%) CNC. The water and UV barrier properties of the composite films showed better performance than those of neat CS film. Results suggested that CNC/CS nanocomposite films can be used as a sustainable packaging material in the food industry.

## 1. Introduction

Petrochemical-based polymeric materials are extensively used in various applications ranging from packaging materials to aircraft components. The broad application of these materials had led to severe environmental challenges due to improper disposal in the environment. In this regard, researchers have focused on biodegradable sustainable packaging materials from scientific and technical sectors [1,2,3]. Hence, bio-based packaging materials based on polysaccharides, proteins, and lipids are increasingly used [4,5]. However, the inferior mechanical and poor barrier properties of natural polymers often hinder their application in food packaging. Reinforcing nanofillers are incorporated into biopolymer formulations to effectively enhance their thermal and physio-chemical properties and fabricate high-performance polymer nanocomposite films [6,7,8]. Nanocomposite films can also serve as water and UV light barriers to improve the shelf-life and quality of preserved food products [9].

Chitosan (CS) is a natural cationic polysaccharide containing numerous reactive amino groups (β-1,4-linked glucosamine and *N*-acetylglucosamine) that can participate in several chemical reactions [10,11]. CS is yielded by the partial deacetylation (to varying degrees) of chitin, which is the second most abundant naturally occurring polysaccharide next to cellulose. CS is very useful in various fields, such as food, agriculture, and wastewater treatment [12]. CS can also be applied to prevent the dehydration of meat, which is the most important diet to sustain human life [13]. CS has been recently exploited in biomedical applications owing to its distinctive properties, namely, biocompatibility, biodegradability, mucoadhesion, excellent film forming properties, nontoxicity, and broad antimicrobial activity against fungi and bacteria [14,15,16,17,18,19,20,21]. Thus, CS is a potential candidate for manufacture of edible films for a large set of applications in pharmacy and packaging industries. Nevertheless, the poor mechanical and barrier properties of CS must be enhanced by combining it with appropriate reinforcement fillers [6,15]. Zhang et al. [22] reported that the addition of TiO_2_ nanoparticles into CS film enhanced the mechanical, thermal, and UV-barrier properties of the neat CS film. Barra et al. [6] evaluated the mechanical properties, electrical conductivity, and antioxidant activity of CS bio-nanocomposite films with and without reduced graphene oxide (rGO).

Over the past decade, natural fillers have been widely applied as reinforcing materials to replace synthetic or inorganic fillers for polymer matrices to obtain bio-based nanocomposites. Given the structural (high aspect ratio, dimension, and alignment), mechanical, and thermal properties of cellulose nanocrystals(CNC), their incorporation into polymers opens a new avenue for achieving high-performance polymer nanocomposites [23]. Cellulose is the most abundant renewable biopolymer on earth and is commonly found in the primary cell wall of green plants [24,25]. Needle-shaped crystalline CNC can be obtained through acid hydrolysis of cellulose fibers. CNC normally possesses an average length of 100 nm–2 µm and a diameter of 5–20 nm [26]. Moreover, the flocculation of CNC depends on the type of hydrolysis [27]. Sometime, isolated CNC does not flocculate with water because of their electrostatic repulsion characteristic on the surface. Thus, CNC suspensions are stable for several months.

CNC has been successfully exposed to reinforce polymers due to the establishment of percolating network that connects well-distributed CNC by hydrogen bonds within polymer matrices. Polymer matrix incorporated with CNC reinforcing filler exhibits superior performance, such as barrier, thermal, and mechanical properties, allowing the manufacturing of next-generation polymer-based nanocomposites [5,28,29,30]. CNC has been widely blended with CS because of the excellent merged properties of CS (biocompatibility, antimicrobial, and tear-resistance) and CNC (good barrier, thermal, mechanical, and high aspect ratio) in CNC/CS nanocomposites. Khan et al. [31] examined the effect of CNC loading on the barrier and mechanical properties of CS-based nanocomposite films; they found that the incorporation of 5 wt.% CNC in CS enhanced the tensile properties of the biodegradable composite films. Salari et al. [5] recently investigated the effect of adding bacterial CNC and silver nanoparticles on the thermal, mechanical, and antimicrobial activity of CS-based composite films. Mujtaba et al. [30] studied the CNC/CS-based composite films and reported that adding 20 wt.% CNC enhanced the mechanical properties of the prepared film. Ma et al. [29] examined CS composites reinforced with modified CNC as cellulose spheres for food packaging applications; these composites exhibited reduced water vapor permeation (WVP) and improved thermal/mechanical properties. 

Among known CS-based composites, CNC/CS nanocomposite is widely investigated [31,32,33,34,35] over the past years. To the best of our knowledge, the UV-barrier properties of CNC/CS bionanocomposites have not been investigated yet. The current study aims to determine the influence of modified CNC on the physicochemical properties of biodegradable/CS-based nanocomposite films intended for use as edible food packaging with improved UV barrier and mechanical properties. The structure, morphology, and thermal/mechanical properties of the fabricated films were evaluated and compared. The water solubility, water absorbency, photo-barrier, and biodegradability properties were also highlighted in this work. Results are predicted to open up a green avenue toward designing and fabricating fully bio-based and high-barrier materials for sustainable food packaging applications.

## 2. Materials and Methods

### 2.1. Chemical and Reagents

CS (molecular weight = 350,000 gmol^−1^; the deacetylation degree = 90%) was purchased from Sigma Aldrich (Tapei, Taiwan). Cellulose microcrystal (CMC) (size 5 μm) was supplied by JRS, Paris, France. Sulfuric acid and glacial acetic acid were obtained from J.T. Baker, Phillipsburg, NJ, USA. Phosphotungstic acid (staining agent) was used as received from Sigma Aldrich (Taoyuan, Taiwan). Deionized water (DIW) was used in the entire study.

### 2.2. Fabrication of CNC from CMC

Sulfuric acid hydrolysis was adopted using a previously reported procedure to extract CNC from a commercial CMC [7]. Oven-dried CMC powder (ca. 3 g, at 70 °C for 12 h) was hydrolyzed with sulfuric acid solution (20 mL, 64 wt.%) at 48 °C for 1 h with continuous mechanical stirring. After hydrolysis, the resultant suspension was poured into 200 mL of cold DIW (<5 °C) to stop the reaction. The reaction was further continued for 1 h with continuous mechanical stirring and terminated by adding 200 mL of cold DIW. The supernatant was eliminated after the completion of the reaction. The decanting step was repeated at least twice for 24 h. The sediment obtained after removing the supernatant was collected and washed by centrifugation at 10,000 rpm for 5 min at room temperature. Centrifugation was performed two to three times before the product was dialyzed in millipore water. The millipore water was frequently replaced until pH of 7 was obtained. The suspension was subjected to ultra-sonication (DELTA Ultrasonic Cleaner DC300H, Delta Electronics, Taoyuan, Taiwan) for 1 h at an operating frequency of 40 kHz and power dissipation of 300 W to avoid the aggregation of unfractionated cellulose nanoparticles. The product was freeze dried and called CNC. With help of Image J software (LOCI, University of Wisconsin, Madison, WI, USA; applying on TEM), the length and diameter of CNC was found 100–500 nm and 5–30 nm, respectively. The obtained CNC yield was ca. 65%.

### 2.3. Synthesis of CNC/CS Nanocomposite Films

CS nanocomposites reinforced with CNC were prepared by solution casting. CS solution (1% w/v) was prepared by dissolving CS flakes into 2% (v/v) glacial acetic aqueous solution and mechanically stirred for 5–7 h. The prepared and formulated CNC (2, 4, 6, and 8 wt.%) was dispersed in 50 mL of DIW, magnetically stirred for 15 min, and sonicated in an ultrasonic bath for 30 min. The obtained CNC water suspension was added to the CS solution and stirred for 1 h at room temperature. The resulting dispersion was degassed under vacuum suction, casted on a petri dish, and dried in an oven at 50 °C for 12–15 h until the solvent was completely evaporated and a self-standing film was attained. The dried CNC/CS nanocomposite films with 0, 2, 4, 6 and 8 wt.% loading of CNC were named as CS0, CS2, CS4, CS6, and CS8, respectively. The procedure regarding the fabrication of CNC/CS films is presented in Figure 1.

### 2.4. Conditioning

Prior to running other tests, prepared films were conditioned at 25 ± 1 °C and 65% ± 2% RH for 48 h in a desiccator. This thermal condition was in the acceptable range of ISO 7730. All the studies were done in triplicate for confirming the reproducibility.

### 2.5. Thickness

The thickness of the films (0.020–0.021 mm) was measured using hand-held Tec lock dial thickness gauge (SM-112, TECLOCK Co., Ltd., Tokyo, Japan) and six random measurements for each film were taken.

### 2.6. Fourier-Transform Infrared Spectroscopy (FTIR)

Fourier-transform infrared spectrometer (TENSOR 27, Bruker, Karlsruhe, Germany) was used for FTIR of samples. Spectra were recorded at wave numbers ranging from 500–4000 cm^−1^. 

### 2.7. Transmission Electron Microscopy (TEM)

The method of sample preparation for TEM was followed by recently published paper [36]. The images of CNC were observed by a JEOL (JEM-1230 electron microscope, JEOL Ltd., Tokyo, Japan) with an accelerating voltage of 200 kV. 

### 2.8. Scanning Electron Microscopy (SEM)

The dispersion of CNC in the chitosan matrix and the particle sizes were observed with a field emission scanning electron microscopy (FESEM; SEM; Jeol JSM-7500F, JEOL Ltd., Tokyo, Japan) operated with an acceleration voltage of 18 kV. All the specimens were sputter-coated with gold before examination.

### 2.9. Optical Microscopy (OM)

To delineate the morphology and dispersion status of CNC within the matrix phase, OM (Olympus BX-50, Olympus Corporation, Tokyo, Japan) was performed at room temperature. 

### 2.10. X-ray Diffraction (XRD)

XRD measurements were performed by an X-ray diffractometer (D2, Bruker, Karlsruhe, Germany) with CuK_α_ radiation source (λ = 0.154 nm) operating at 40 kV and 40 mA. A scattered radiation was recorded at an ambient temperature in the 2θ range of 10–40° using a step interval of 0.02° and scan speed of 1°/min. The degree of crystallinity was measured by Segal’s empirical method with the following equation [37]:(1)$$\mathrm{CI}\left(\%\right)=\frac{I_{002}-I_{\mathrm{am}}}{I_{002}}\times 100,$$
where I_002_ is the intensity value for the crystalline cellulose (2θ = 22.5°) and I_am_ is the intensity value for the amorphous cellulose (2θ = 16.3°).

### 2.11. Thermogravimetric Analysis (TGA)

A thermogravimetric analyzer (TA Q50, TA Instrument, New Castle, DE, USA) was carried out to analyze the thermal stability of the films. The measurement were perused with a temperature range from room temperature to 700 °C at heating rate of 10 °C/min under both nitrogen and air atmospheres. 

### 2.12. Opacity and UV Visibility

The opacity of prepared films was determined by measuring the film absorbance at 600 nm using UV spectrometer (JASCO, V-650, Tokyo, Japan) and calculating by the equation:(2)$$\mathrm{Opacity}={\mathrm{Abs}}_{600}/d,$$
where, Abs_600_ is the value of absorbance at 600 nm and d is the film thickness (mm). 

### 2.13. Water Absorbency (WA) of the Films

Water absorbency (WA) of prepared film is examined in terms of percentage swelling ratio. The details about the procedure and condition for WA were followed by our recently published paper [7].
(3)$$\mathrm{Swelling}\;\mathrm{Ratio}\;\left(\%\right)=P_{s}=\frac{W_{s}-W_{d}}{W_{d}}\times 100,$$
where, $W_{d}$ and $W_{s}$ is the weight of dry and swollen samples, respectively. 

### 2.14. Equilibrium Moisture Content (EMC)

The equilibrium moisture content (EMC) of the prepared films was calculated by measuring the weight loss of films pieces (2 × 2 cm^2^) upon dried in a hot oven at 102 ± 2 °C to reach the equilibrium weight $\left(W_{\infty }\right)$ with its initial (dry) weight $\left(W_{o}\right)$, as follows:(4)$$\mathrm{EMC}=\frac{W_{\infty }-W_{o}}{W_{o}}\times 100.$$

### 2.15. Film Water Solubility (FWS)

FWS was deliberated by recently published paper [7,38]. The samples (2 × 2 cm^2^) were dried at 27 °C for 24 h to determine the initial dry weight (W_i_). Then, the film were immersed in 20 mL of deionized water with mild shaking, removed, and then dried at 40 °C for 24 h to determine the undissolved final dry weight (W_f_). FWS was calculated using the following equation [39]:(5)$$\mathrm{FWS}=\frac{W_{i}-W_{f}}{W_{i}}\times 100.$$

### 2.16. Contact Angle

A Drop Shape Analysis System (model DIGIDROP GBX instrument, GBX, Romans sur Isere, France) was operated for contact angle measurements. The hydrophilic nature of the prepared films was evaluated by deionized water.

### 2.17. Water Vapor Permeation (WVP)

According to ASTM-E96/E96-05 Standard method, the WVP of CS nanocomposite films was studied. The details about the procedure for measuring WVP of samples was followed by Yadav et al. [7]. In each test petri dish, 20 mL of distilled water were added, leaving a distance of approximately 1 cm between the water surface and the film. The film samples were sealed to the dish mouth by a water-resistant sealant. The petri dishes were conditioned in a humid chamber (Giant Force, Tapei, Taiwan) at 25 °C to ensure 65% RH, and weight was noted a certain interval of time. The provided Equation (6) was used to calculate the WVP data.
WVP (g m^−1^s^−1^ Pa^−1^) = (w/t) γ (A)^−1^ (Δp)^−1^.(6)
where, w is the weight loss of the Petri dishes (g), γ is the film thickness (m), A is the cross-section area of the film (m^2^), t is the time (s), and Δp is the vapor pressure difference. 

### 2.18. Mechanical Properties

The mechanical properties of the films (according to ASTM D638) were performed using a Gotech AI-3000 system (Gotech Testing Machines Inc., Taichung, Taiwan) at room temperature (24 ± 2 °C). Crosshead speed was set at 10 mm/min for all the specimens. The pull tests of the films in terms of tensile strength and elongation at break (%) was calculated by published papers [36].

### 2.19. Soil Burial Test

Soil burial tests for film biodegradation characteristics were conducted according to previously reported method described by Martucci et al. [40]. The samples were cut into rectangular parts (1 cm × 1 cm × 0.002 cm) and dried in an oven at 100 °C for 5 h prior to weighed ($W_{i}$). A total amount (30 g) of garden soil was poured into a plastic pot up to a thickness of approximately 1 cm. Then, the films were buried under soil at a depth of 0.5 cm from the soil surface. The assay was performed at 25 °C and 30% RH by using a humid chamber (Giant Force, Tapei, Taiwan). Samples were taken from the soil at different times and cleaned carefully with tissue paper. Subsequently, the samples were dried in an oven at 105 °C for 6 h and weighed ($W_{t}$) to determine the percentage of weight loss using the following equation:(7)$$\mathrm{Weight}\;\mathrm{loss}\left(\%\right)=\frac{W_{i}-W_{t}}{W_{i}}\times 100.$$

## 3. Results

### 3.1. FTIR

The FTIR spectra of CS, CMC, CNC, and CS4 are presented in Figure 2. The FTIR spectra of CNC confirmed that a sharp peak at 3350 cm^−1^ was found due to the OH vibrations in hydrogen bonds [41]. The absorption bands between 2800 and 3000 cm^−1^ originated from C–H stretching and bending vibrations [42]. The other absorption bands were assigned at 567 (O–H out of plane bending vibrations), 1058 (C–O stretching at C-3 position), and 1163 cm^−1^ (C–O–C stretching motion) [43]. Aside from these peaks, the CNC showed two more peaks at 1645 and 1240 cm^−1^, confirming the acidification of the CMC was successfully performed [44]. Furthermore, the abundant oxygen functional groups rendered the CNC hydrophilic, which improved their solubility in water. When CNC was added to CS, the peaks at 3392–3405 (NH_2_ stretching vibration), 1539–1542, 1341–1335, 1029–1032, and 1063–1072 cm^−1^ sharpened with increased intensity, indicating that CNC interacted with CS through hydrogen bonds [23]. Furthermore, the appearance of two peaks at 2916 (symmetric C–H vibration) and 2856 cm^−1^ (symmetric and asymmetric C–H vibration) in CNC/CS confirmed the bonding of CS and CNC.

### 3.2. XRD

The physical interaction between biopolymer and nanofiller can be understood based on degree of crystallinity. XRD analysis was conducted within the 2ϴ range of 10–40° on CMC, CNC, CS0, CS2, CS4, CS6, and CS8 samples. As shown in Figure 3, CMC showed four crystalline peaks at 15.3°, 16.5°, 22.6°, and 34.7°. However, after acid treatment of CMC, CNC exhibited peaks at 15.6° (101), 16.4° (101), 23.0° (200), and 34.6° (400) [7,45]. The calculated crystallinity index (CI) values of CMC and CNC were ca. 40% and 50%, respectively, suggesting that structural change by the acid treatment of CMC was responsible for the increased CI of CNC. For CS, two main peaks appeared at 10°(002) and 20°(101) due to its crystalline nature. The CI of chitosan film was found 35%, which was agreed with previously published paper [46]. With the addition of CNC to the CS matrix, the mechanical and physical properties of the films were changed due to the crystalline morphology, crystal forms, and crystallite size of CNC [47]. In the case of blends the diffraction peaks for both chitosan and CNC were sifted to a higher value with increasing CNC content, thus the peaks from 10° was shifted to 12.05°, 15.6° at 15.7°, 16.4° at 16.7°, 20° is combined with that from 23° of CNC and is shifted at 22.75°, and the signal at 34.6° at 34.65°. These shifting induce an increasing in the lattice distance with increasing the CNC content. The incorporation of the rigid molecules of CNC in the chitosan matrix induced an orientation of the polymeric molecules and favored the formation of hydrogen bonds between the polymeric chains.

### 3.3. Microstructure and Dispersibility of CNC

The FESEM image of aggregated needle-shaped CNCs in bundle form is presented in Figure 4a. The bundles of CNC were easily dispersed in water due to their modified nature. As shown in the TEM image of Figure 4b, the length and width of the obtained needles were 100–500 and 5–30 nm, respectively. The optical microscope (OM) images of the CMC, CNC, CS0, CS2, CS4, CS6, and CS8 samples in dry form at 20× magnification are presented in Figure 5. The bundles of CMC and CNC are presented in Figure 5a,b, respectively. As predicted from Figure 5c, a smooth surface of CS0 film was observed. The surfaces of CS2 and CS4 revealed a uniform and dense arrangement due to the small size and homogeneous distribution of CNC in the films. However, CS films with high CNC content (cf. CS6 and CS8) had rough surfaces. Thus, a stiff break with a high loading of CNC is recommended. The cross-section micrographs of CS0 and CS4 are presented in Figure 4c,d. Figure 4d shows the increased smoothness of CS upon further loading of CNC, suggesting the effect of the distribution of CNC particles. The tensile fractured micrograph of CS0 and CS4 showed roughness due to the presence of broken fibers (Figure 4e,f). No phase separation was found, indicating good adhesion between the CS and the CNC. Structural studies confirmed that the incorporation of CNC into the CS matrix resulted in a homogeneous composite, in which the individual components are connected by hydrogen bonds [48]. Similar findings were reported by Garcia et al. [49] and Yadav et al. [7]. The uniform and homogeneous distribution of CNC in the nanocomposite film is responsible for the enhanced physico-mechanical properties of the CS based film.

### 3.4. Thermal Stability of Films

Biomaterial-based packaging films are extensively used in the food industry due to their short processing time and high yield with low production cost. However, the temperatures reduce film stability due to degradation [50]. Therefore, the thermal behavior of the fabricated films must be evaluated. Here, TGA was performed to evaluate the effect of CNC on the degradation behavior of CNC/CS composite films (Figure 6). Figure 6a,b illustrates the TGA/DTGA curves of all of the samples scanned under nitrogen environment. As shown in Figure 6a, neat CMC and CNC exhibited similar decomposition curves. CNC possessed slightly higher degradation temperature than CMC. Two-staged degradations were observed during the thermal scanning of the samples. A small weight loss occurred in the first stage of degradation at ca. 100 °C, which could be attributed to the evaporation of absorbed water in the sample. The major weight loss in the second stage of degradation at 300–400 °C may be due to the concurrent cellulose degradation processes, such as dehydration, depolymerization, and decomposition of the glycosidic bonds of CMC/CNC [7].

Neat CS and CNC/CS with different combined ratios showed multiple degradation curves. The composites showed slightly higher degradation curves than the neat CS. After CNC inclusion, the thermal stability of CNC-reinforced CS composite films showed marginal improvement due to interactions between CNC and CS matrix [51]. Figure 6b depicts the derivative TGA (DTG) curves of the samples. Neat CMC showed a lower peak temperature (T_max_, temperature at maximum weight loss rate) than CNC, as illustrated in the inset of Figure 6b. The neat CS and CS with different weight ratios of CNC exhibited three peak temperatures (T_maxI_, T_maxII_, and T_maxIII_). T_maxI_ (first stage) occurred at the range of 40–100 °C, which corresponds to a weight loss of ca. 14% and is attributed to the evaporation of residual acetic acid [52] and physical trapping of water and/or weakly hydrogen-bonded to CS molecules. The second stage (T_maxII_) of weight loss at below 200 °C could be assigned to the initial degradation of the polymer and/or release of atypical, strongly hydrogen-bonded water. The predominant (final stage) thermal degradation peak occurred within 250–400 °C (T_maxIII_ at 292 °C), during which a 48% decline of the CS mass was detected. This finding could be due to the depolymerization of CS chains and the decomposition of pyranose rings [53] through dehydration and deamination, followed by final ring-opening reaction [54].

TGA was also performed in air environment to evaluate the thermal oxidation behavior of CS and CNC/CS composite films (Figure 6). The degradation behavior of CS and CNC/CS under oxidized conditions was substantially different from that under nitrogen atmosphere. As shown in Figure 6c, the samples exhibited slightly lower degradation temperature than their corresponding nitrogen environment-scanned samples because of auto-oxidation [54]. The neat CNC exhibited higher degradation temperature at 50 wt.% loss compared with neat CMC. All of the CS samples (CS and CNC/CS) showed similar degradation behavior appeared within 200–350 °C with 57% weight loss occurring at almost 292 °C as observed in nitrogen. The main difference between nitrogen- and air-atmosphere-scanned samples is the additional decomposition peak (T_maxIV_) that occurred above 400 °C, as depicted in Figure 6d. The strong exothermic effects in the presence of oxygen indicate the occurrence of efficient oxidation, followed by further decomposition of oxidized CS. These stages with high weight loss are attributed to very efficient chain scission with the formation of volatile degradation products [54]. All T_max_ values and residue wt.% of the samples scanned under nitrogen and air environments and temperatures at 50 wt.% loss (T_d50_) are listed in Appendix A. All of the composites films scanned under nitrogen and air environment exhibited T_50%_ values above 285 °C. Hence, these composite materials are regarded as stable at high temperatures and have potential as packaging materials.

### 3.5. Water Absorbency of CNC/CS Film

The water absorbency in terms of percentage swelling ($P_{s}$) is an important property for a film when considering its application in food packaging. The $P_{s}$ values of the fabricated films were evaluated and compared (Table 1). Neat CS film exhibited $P_{s}$ ca. 818.57%. The presence of CNC considerably decreased the $P_{s}$ of the CS film. For example, when 2 wt.% CNC (cf. CS2) was loaded in the CS matrix, the $P_{s}$ value decreased from 818.57% to 537.85%. The $P_{s}$ values of the composites further decreased (up to $P_{s}$ value ca. 392.85%) with increasing CNC content. This decrease may be dependent on the CNC crystalline nature [55]. Some researchers [31] reasoned that the 3D network of CNC moderated the flexibility of CS molecular chains by reducing the number of –OH, thereby blocking the infiltrate paths of water molecules and confirming that CNC decreased the water absorbency of CS. Similar observations were previously reported [56,57,58].

### 3.6. Equilibrium Moisture Content (EMC)

The EMC values of CS0, CS2, CS4, CS6, and CS8 films were 34.17%, 29.33%, 27.54%, 25.81%, and 24.27%, respectively. As shown in Table 1, the samples containing CNC (cf. CS2, CS4, CS6, and CS8) showed lower EMC than CS. The decrease in the EMC value could be due to the less availability of free hydroxyl groups of the matrix that participated in hydrogen bonding to CNC. Ojagh et al. [59] and Fernandes et al. [60] reported similar results. Furthermore, the EMC results of the samples were 24.27–34.17% lower than that those (13–18%) reported by Dehnad et al. [32].

### 3.7. Film Water Solubility

The FWS values of CS0, CS2, CS4, CS6, and CS8 films were 35.70, 33.85, 31.91, 28.20, and 26.55, respectively. As shown in Table 1, the FWS of the samples decreased from 35.70 to 26.55 with increasing CNC loading (0–8 wt.%). This phenomenon can be explained on the basis of 3D network structure of CNC, which protected the movement of polymers to water, resulting in the reduced FWS of the samples [58,61]. 

### 3.8. Contact Angle (CA) of Nanocomposite Film

The CA values of CS0, CS2, CS4, CS6, and CS8 films were 89.91°, 101.70°, 105.55°, 113.62°, and 118.90°, respectively. As shown in Table 1, CS0 and CS8 showed the highest hydrophilicity (lowest hydrophobic) and the lowest hydrophilicity (highest hydrophobic) among all of the tested films, respectively. The increase in the contact angle may be due to the hydrophobic nature of CNC, could be beneficial for the water resistance of CS food packaging films. Mao et al. [56] reasoned that CS and CNC joined by hydrogen bonds improved the hydrophobic nature of the composite films. The work of adhesion (WA or W_12_) is the work applied to separate two adjacent phases, namely, 1 and 2. W_12_ depends on the contact angle and surface tension of the liquid and is the energy that is released during wetting. On the basis of Young–Dupree equation:WA = W_12_ = (1 + Cos ϴ) γ(8)
where γ is the surface tension. 

W_12_ can be correlated with the interaction of the filler with the matrix and with the interaction of the filler with a liquid that is comparable with the matrix polymer [62]. The WA values of all samples are shown in Table 1. CS0 and CS8 showed the highest (72.99) and lowest (37.62) work of adhesion, respectively. The decrease in the WA of the composite films could be due to the weak dispersion of CNC in the CS matrix [63].

### 3.9. WVP

Table 1 shows the WVP data of CS0, CS2, CS4, CS6, and CS8 films. The WVP of the composite films (2.91–2.41 × 10^−11^ gm^−1^ s^−1^Pa^−1^) was lower than that of CS (3.83 × 10^−11^ gm^−1^ s^−1^Pa^−1^) film. The reduction can be explained by the physical barrier to the passage of water provided by CNC. Numerous studies [31,33,64] reported a similar behavior. The WVP of mango puree films improved significantly with the addition of CNC [65]. Paralikar, Simonsen, and Lombardi [66] also reported the reduction in the WVP of PVOH films due to the addition of 10% (w/w) CNC.

### 3.10. Opacity and UV Visibility

The study of opacity governed the features of the transparent food packaging films. Opacity is a type of degree that does not allow light to pass through a material. The lower the opacity is, the higher the amount of light that can pass through the material will be. The opacity data of the reported films are listed in Table 2. Neat CS exhibited an opacity ca. of 0.916 mm, which was evidently lower than that of the composites. All composites (cf. CS0-8) showed evidently higher opacity values due to the high loading of CNC, demonstrating that the composites can evidently hinder light to pass through the film compared with controlled CS. The UV absorbance graphs of CS and its composite with CNC (cf. CS0-8) is presented in Figure 7A. The UV light absorption increases with increasing CNC due to presence of the 3D structure of CNC. Furthermore, CNC blocked UV light, further creating films resistant to UV wavelengths. Zhang et al. [67] found that CNC-reinforced PVC thin film reduced UV absorption.

The OM images presented in Figure 5 demonstrate the surface roughness of CS0, CS2, CS4, CS6, and CS8 films. The difference in the surface nature between CS0 and CS is due to the presence of CNC (cf. CS2, CS4, CS6, and CS8) films. The CS0 films showed more uniform surface than CS containing CNC, and the surface uniformity decreased with increasing CNC content. Furthermore, the improvement in the transmittance of films was ascribed to the increase in the surface smoothness (low roughness) of the film due to the decrease of surface light scattering [68].

The transparency of the CS0, CS2, CS4, CS6, and CS8 films was studied using UV/visible light spectroscopy. Figure 7C presents the transmittance of light at 800 nm. The transmittance of the CS nanocomposite films significantly decreased from 94.62% to 61.17% with an increase in the CNC loading (0–8 wt.%). This result may be due to the agglomeration of CNC in the CS matrix. Li et al. [69] also reported that the transmittance of composite films rapidly decreased from 89% to 62% with further CNC loading. The digital images of the obtained transparent films are shown in Figure 7B. The order of transparency is as follows:CS0 > CS2 > CS4 > CS6 > CS8;95.08 > 79.22 > 69.81 > 62.32 > 56.11.

The transmittance spectra clearly show that the transmittance decreased with increasing CNC content because of light scattering [7]. The blocking effect [70] of CNC for the used films was determined using the following formula:(9)$$\mathrm{Beff}=\frac{\mathrm{TCS}-\mathrm{TCNC}/\mathrm{CS}}{\mathrm{Percentage}\;\mathrm{of}\;\mathrm{CNC}\;\mathrm{with}\;\mathrm{respect}\;\mathrm{to}\;\mathrm{the}\;\mathrm{CS}}$$
where T_CS_ and T_CNC/CS_ are the transmittance of CS film and the CNC/CS nanocomposites, respectively. Figure 7D shows the B_eff_ of the CNC at 300, 700, and 750 nm (UV-B, UV-A, and visible regions, respectively). Based on the results, the addition of CNC conferred CS with high barrier properties, resulting in enhanced UV light resistance. 

### 3.11. Mechanical Properties

Enhanced mechanical properties are necessary for edible films developed in the food industry because they can hold integrity during processing, handling, and shipping [71]. Accordingly, the mechanical performance of CS and CNC/CS films with various amounts of CNC loading were evaluated by typical tensile experiments. Values were determined in terms of Young’s modulus (YM), tensile strength (TS), and elongation at break (EB). The used films were rectangular with a size of 10 mm × 60 mm × 0.02 mm for tensile tests. The tensile tests of the CS0, CS2, CS4, CS6, and CS8 films (in accordance with ASTM D638) were conducted at a crosshead speed of 10 mm/min by using a Gotech AI-3000 system. The YM, TS, and EB were taken to be the average value of the five tests for each composition. As shown in Table 2, a small amount of CNC loading significantly improved the tensile properties. The TS values of CS2 (79.3 ± 2.6 MPa), CS4 (104.7 ± 1.3 MPa), CS6 (101.4 ± 1.8 MPa), and CS8 (99.6 ± 1.5 MPa) were higher than those of CS0 (75.2 ± 1.6 MPa). Thus, the TS values of CS2, CS4, CS6, and CS8 were 5.45%, 39.23%, 34.84%, and 32.45% higher than that of CS0. The improvement in TS may be due to the strong hydrogen bonding between the CNC and the CS matrix phase [72]. Furthermore, the TS of the films decreased from 104.7 ± 1.3 MPa to 101.4 ± 1.8 MPa when the loading amount of the CNC was increased from 4 to 6 wt.%. The value further decreased to 99.6 ± 1.5 MPa with further increase in the loading amount of CNC at 8 wt.%. This finding may be attributed to the agglomeration of CNC in the CS matrix [73]. Khan et al. also found similar trends [31,58]. Numerous tensile observations [69,74,75] for packaging with low loading of nanofillers in various commercial biopolymer were examined.

The TS of the CS control film (75.2 ± 1.6 MPa) obtained in this study was similar to that of films in previous reports, such as ca. 72.0 [51], 79 [76], 21.07 [5], 46.50 [34], 85.0 [55], 32.9 [64], and 47.68. MPa [33]. The TS values of films were comparable to those of plastic-based films, such as films based on LDPE (8–10 MPa), HDPE (19–31 MPa), EVOH (6–19 MPa), PCL (4 MPa), PS (31–49 MPa), PLA (45 MPa), PVC (42–55 MPa), and PP (27–98 MPa) [50]. Li et al. [69] found that the TS of the composite films increased from 85 to 120 MPa, with an increase of CNC content from 0 to 15 wt.%. Yu et al. [77] studied that the TS of the PVA/CS biodegradable films with 0.6 wt.% was as high as 44.12 MPa and improved by 45% through hydrogen bonds between silica and PVA or CS. Barra et al. [6] reported that the loading of (50 wt.%) rGO in CS matrix is increased by two and six times that for TS and YM, respectively. Some other papers reported less improvement in the TS and EB (%) of CS on the loading of Na^+^MMT/CS, Cloisite 30B, and silver nanoparticles [64].

As presented in Table 2, the YM values of CS2, CS4, CS6, and CS8 were significantly improved by 38.8%, 78.6%, 72.6%, and 70.0%, respectively, relative to that of CS0. The improved YM values of reinforced CS films may be ascribed to the improved stiffness of the films by the inclusion of CNC. This observation suggests that the added CNC acts as good reinforcing agents to the CS. Hosseini et al. [78] also reported similar results. Tang et al. [79] observed that the CS/gellan gum nanocomposite films with 7% CNC exhibited increased TS and YM by 80.9% and 41.7%, respectively, and decreased EB by 53.1% relative to those of pure CS/gellan gum composite films. Pan et al. [80] observed that after forming the nanocomposite film with (1 wt.%) GO loading, the YM, TS, and EB are increased by 51%, 93%, and 41%, respectively, which are much higher than those of neat CS films. With the addition of only (0.2 wt.%) unzipped multiwall carbon nanotube oxides into the CS matrix, the TS and YM are increased from 69.3 to 142.7 MPa and 2.6 to 6.9 GPa, showing increases of 105.9% and 165%, respectively, as reported by Fan et al. [81].

The EB values of CS0, CS2, CS4, CS6, and CS8 were 21.8% ± 0.5%, 16.8% ± 0.6%, 9.9% ± 0.4%, 9.2% ± 0.5%, and 8.9% ± 0.3%, respectively (Table 2). The EB of the nanocomposite films decreased from 21.8% ± 0.5% to 8.9% ± 0.3% when the loading amount of the CNC was increased from 0 to 8 wt.%. This result may be due to the lesser availability of the motion at the interface of filler–polymer [82]. Recently, similar observations were also found by Barra et al. [6]. The EB of the prepared nanocomposites is comparable to those of films based on PS (2–3%), PVC (20–180%), and PVDC (10–40%). On the basis of improved TS and YM of CNC reinforced samples, we could conclude that these films can be used in the packaging industry. 

### 3.12. Comparison of Obtained Mechanical Performance between the Present Study and Previous Works

As presented in Appendix A, the mechanical behavior of CNC4 was higher than that previously reported [5,30,31,33,34,35,51,55,60,83] suggesting that CNC/CS films can be used in the food packaging industry. 

### 3.13. Biodegradability and Stability of the Prepared Films

The food industry is one of the highest packaging disposal creators; as such, obtaining biodegradable packages for foodstuff is an important requirement to avoid environmental problems. Therefore, researchers have focused on using sustainable biomaterials for food packaging. The biodegradability of food material is well studied by Yabannavar et al. [84]. The word “biodegradable” is used to describe materials that can be degraded by the enzymatic action of living organisms [85,86]. A biodegradable material is a good alternative for traditional non-biodegradable materials due to its recycling behavior. Thus, we studied the biodegradability of CS and CNC-reinforced CS. As presented in Figure 8A, the biodegradability of CS thin film increased up to 70.27% as the burial time increased in the garden soil for 8 days. However, in the case of CNC-reinforced CS films, the rate of degradation decreased due to presence of strong hydrogen bonds between CNC and CS matrix [87]. Therefore, the CNC incorporation in CS matrix decreased the degradation percentage of films by up to 8 wt.% of CNC.

The stability of prepared films in water is presented in Figure 8B. After dipping in water for 30 days, neat CS film was degraded and broken into small pieces [88]. However, CNC-reinforced CS films showed less degradation and remained almost intact. The obtained results revealed that the prepared CS2, CS4, CS6, and CS8 films showed improved stability. Therefore, we can assume that CNC-reinforced CS films can be used as packing materials in the food industry. 

## 4. Discussion

CNC-reinforced CS-based sustainable biocomposite films with enhanced properties were successfully obtained by mixing CNC and CS through solution casting. The needle-shaped CNC was successfully synthesized from CMC by acid-hydrolysis treatment. The modification of the chemical structure of CNC/CS composite was analyzed by FTIR and XRD studies, revealing the formation of hydrogen bonds between CNC and CS. FESEM and OM images confirmed a dispersion of CNC throughout the CS matrix without remarkable agglomerations. TGA results indicated that chitosan and its composites exhibited almost same thermal stability. In the presence of CNC in the nanocomposites, water and UV barrier properties were also improved. FWS and EMC were reduced by 27% and 32%, respectively, after incorporation of only 4 wt.% CNC. The aim of improving UV absorption with the presence of CNC in composites was achieved, and the composites evidently showed lower transmittance compared with control CS. The TS and YM of CS film was improved with the incorporation of CNC, whereas EB decreased with increasing brittleness of the composite films. Moreover, the composites with CNC loading showed lower biodegradation percentage than the control CS due to the presence of strong interaction through H-bonding between CS and CNC. Therefore, CNC/CS bionanocomposite film could be candidates for potential applications in the food packaging field.

## Figures and Tables

**Figure 1 polymers-12-00202-f001:**
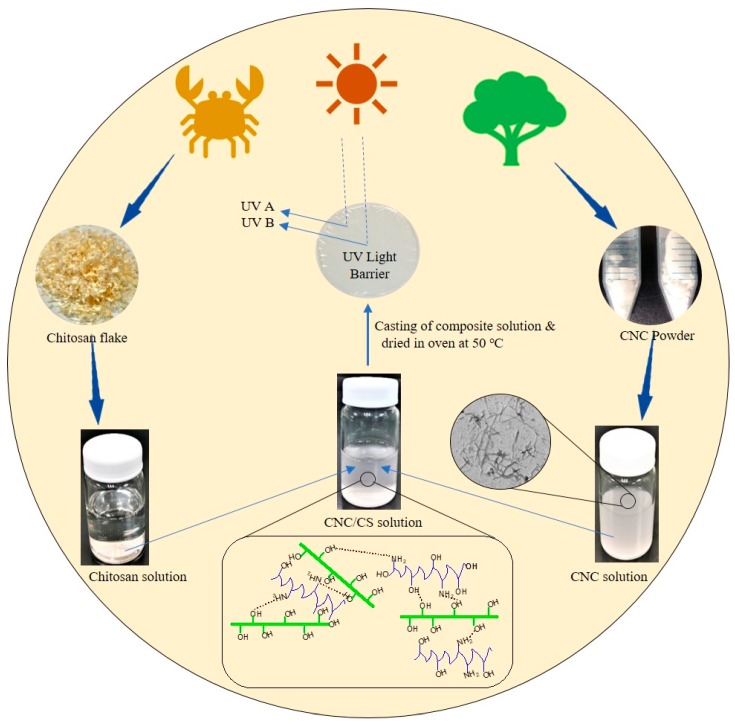
Schematic of preparing cellulose nanocrystal/chitosan (CNC/CS) nanocomposite films.

**Figure 2 polymers-12-00202-f002:**
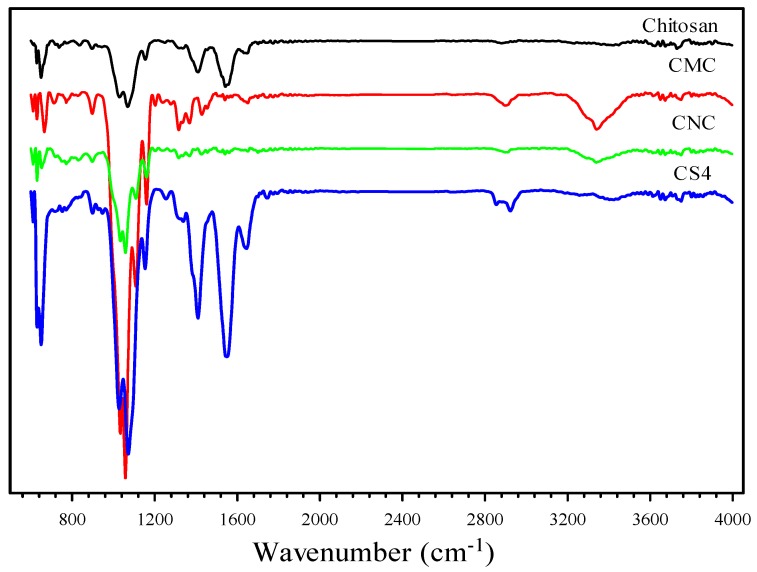
FTIR of CS, CMC, CNC, and CS4 samples.

**Figure 3 polymers-12-00202-f003:**
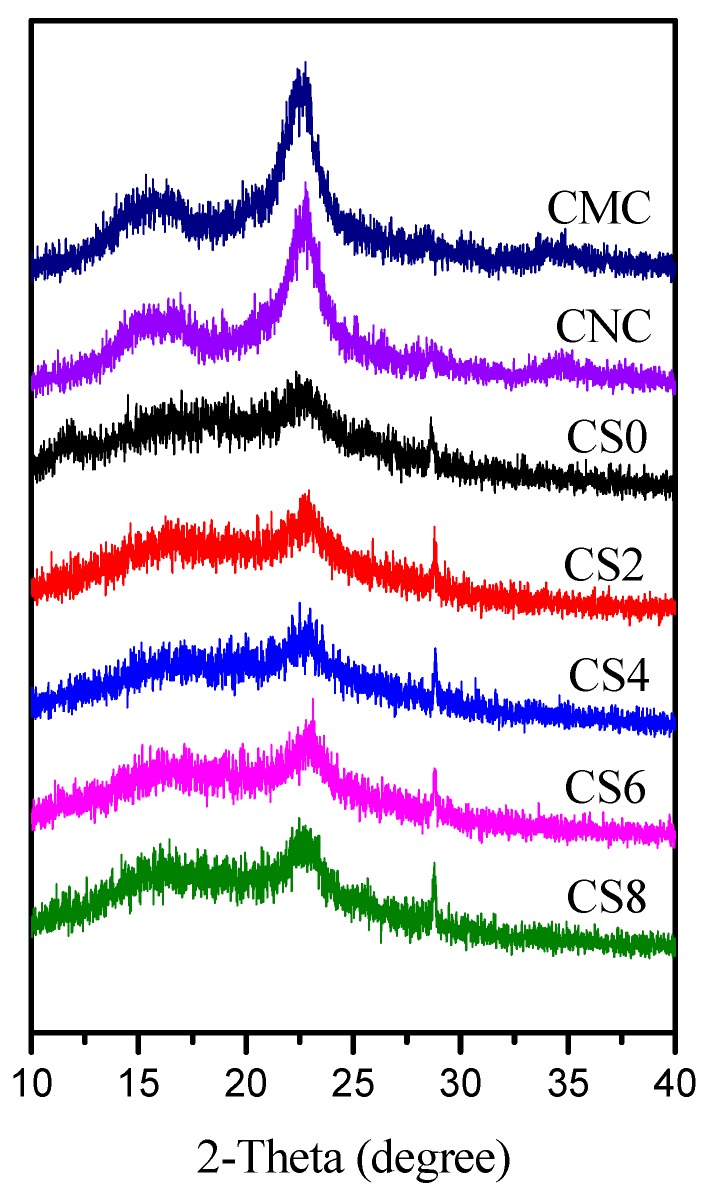
XRD of CMC, CNC CS0, CS2, CS4, CS6, and CS8 samples.

**Figure 4 polymers-12-00202-f004:**
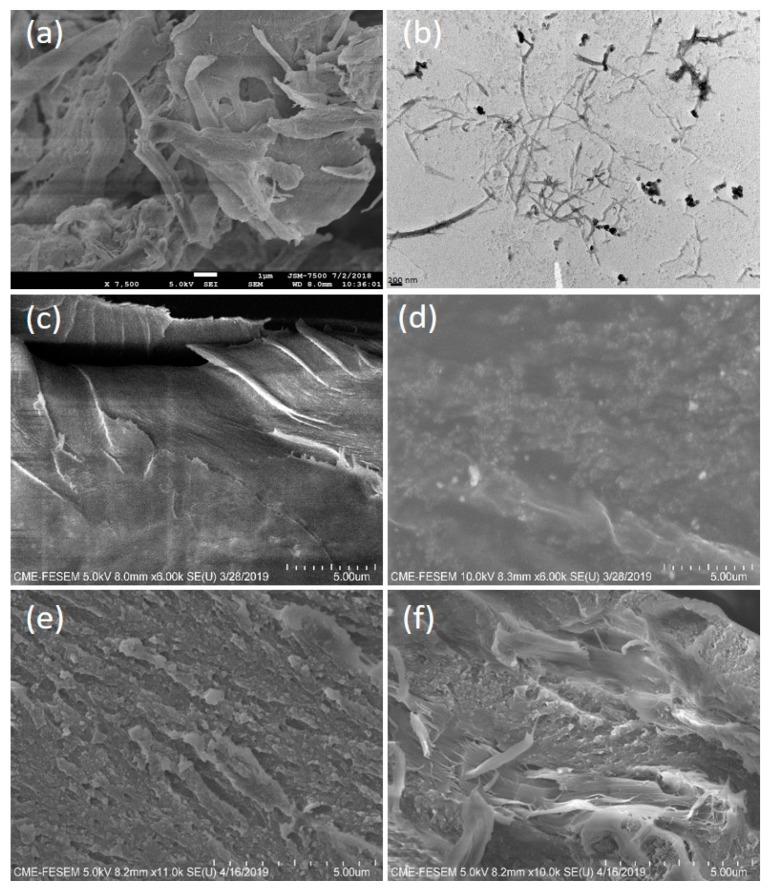
(**a**) Field emission scanning electron microscopy (FESEM) of CNC in powder form, (**b**) TEM image of water dispersed CNC, (**c**) FESEM cross-section image of CS0 films, (**d**) FESEM cross section image of CS4 film, (**e**) FESEM cross-section image of fractured CS0 films, and (**f**) FESEM cross-section image of fractured CS4 film.

**Figure 5 polymers-12-00202-f005:**
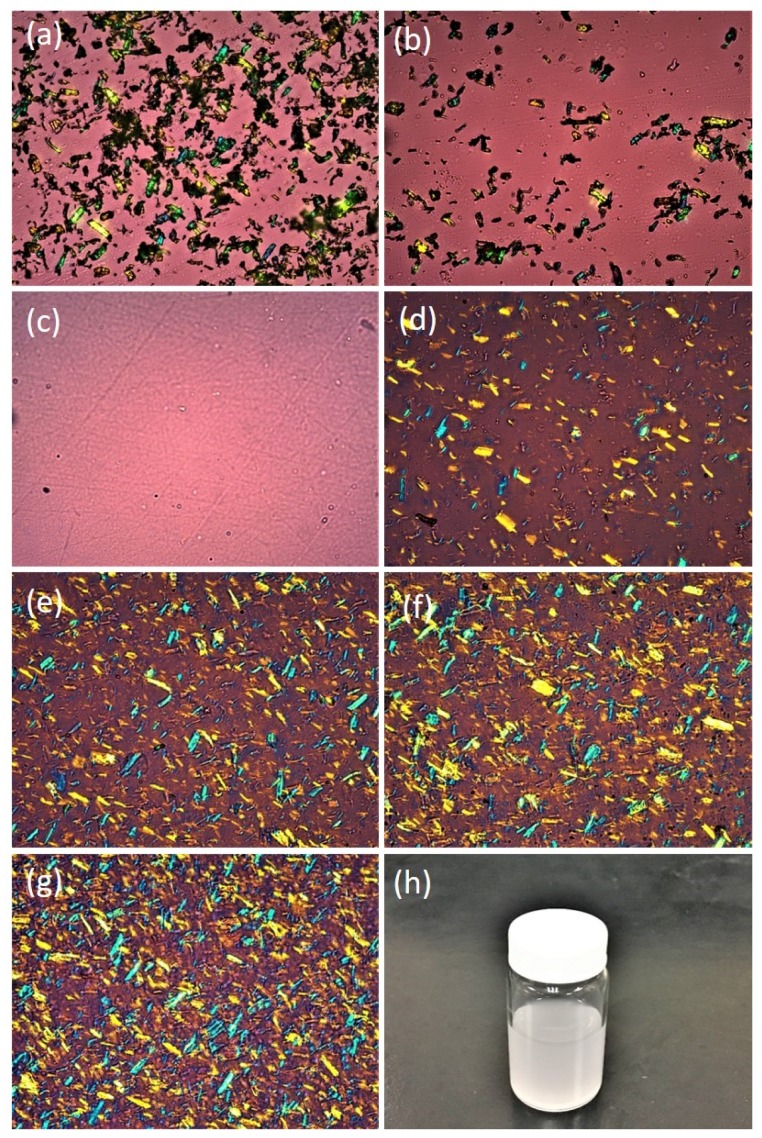
Optical microscope (OM) images of (**a**) CMC in powder form, (**b**) CNC in powder form, (**c**) CS0 film, (**d**) CS2 film, (**e**) CS4 film, (**f**) CS6 film, (**g**) CS8 film, and (**h**) digital image of water dispersed CNC.

**Figure 6 polymers-12-00202-f006:**
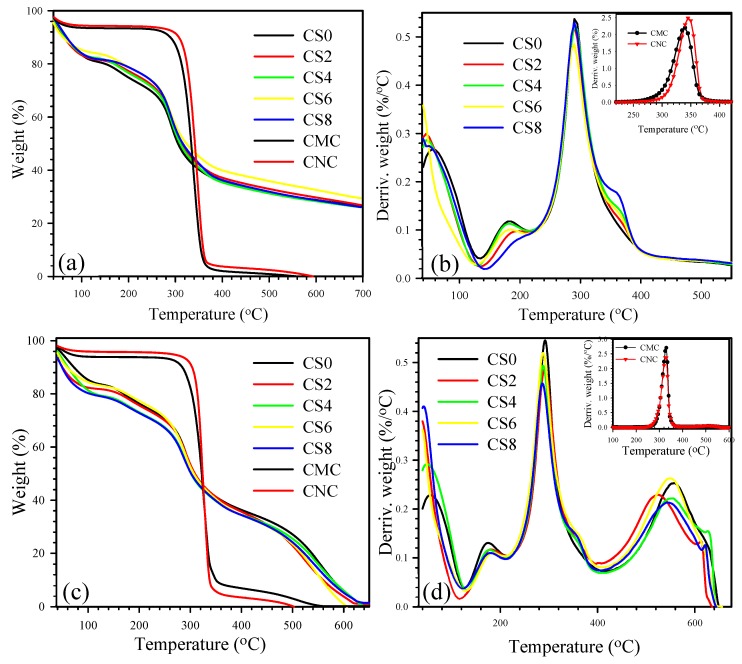
Thermal decomposition curves of CS0, CS2, CS4, CS6, CS8, CMC, and CNC: (**a**) and (**b**) N_2_ environment, and (**c**) and (**d**) air environment.

**Figure 7 polymers-12-00202-f007:**
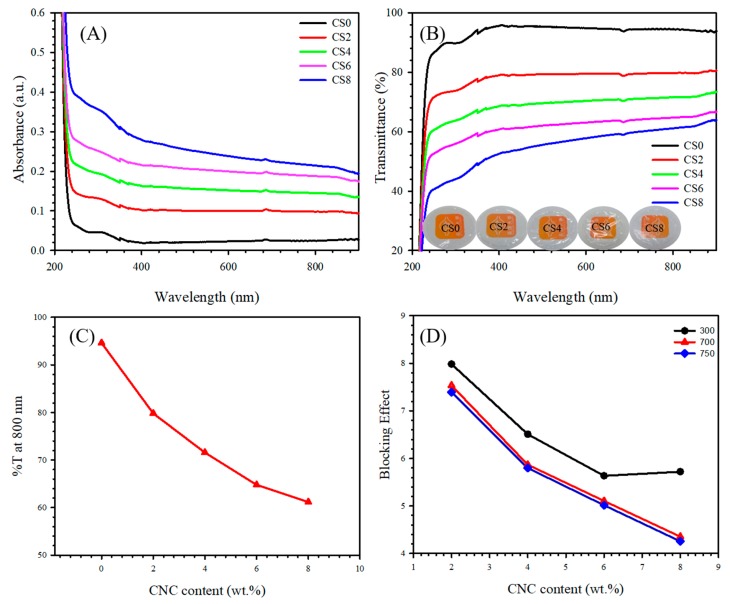
(**A**) UV absorption curve, (**B**) UV transmittance curve, (**C**) optical percentage transmittance (%T) of UV at 800 nm, and (**D**) blocking effect of UV at different wavelengths for CS0, CS2, CS4, CS6, and CS8 films. Inset of Figure (**B**): digital image of fabricated films.

**Figure 8 polymers-12-00202-f008:**
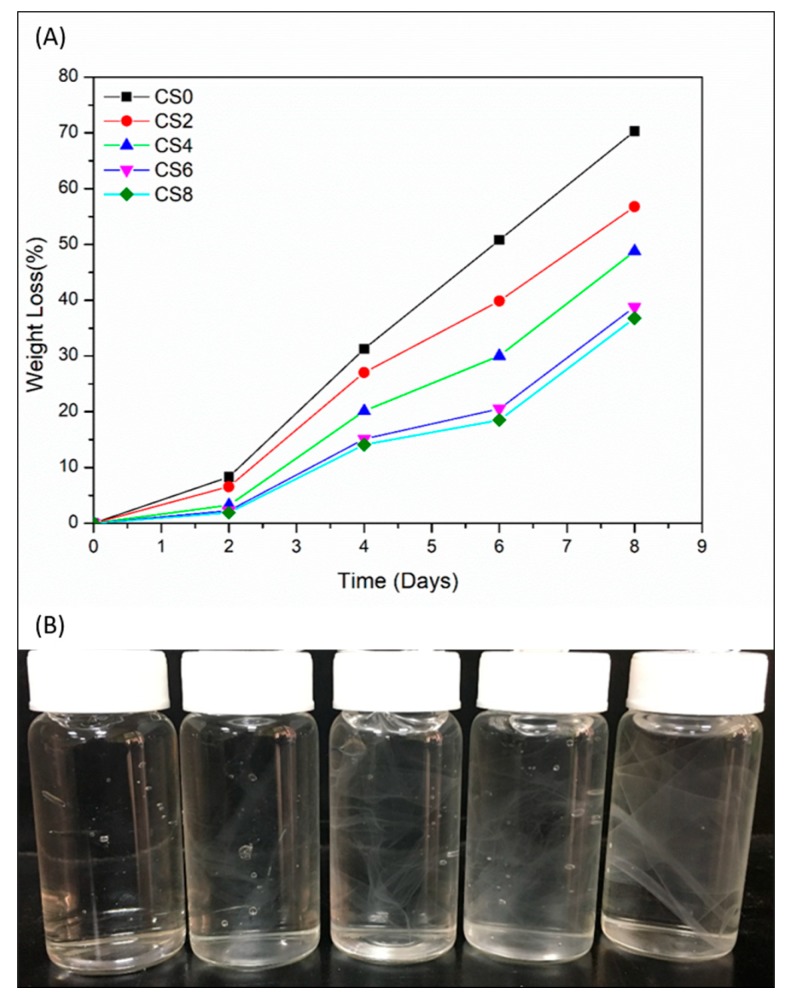
(**A**) Biodegradability and (**B**) stability of CS0, CS2, CS4, CS6, and CS8 films.

**Table 1 polymers-12-00202-t001:** Water barrier properties of the samples.

Properties	Samples				
	CS0	CS2	CS4	CS6	CS8
P_s_	818.57	537.85	436.87	418.51	392.85
EMC (%)	34.17	29.33	27.54	25.81	24.27
FWS (%)	35.70	33.85	31.91	28.20	26.55
CA/◦	89.91	101.70	105.55	113.62	118.90
WA	72.99	58.04	53.41	46.89	37.62
WVP	3.83	2.91	2.72	2.54	2.41

**Table 2 polymers-12-00202-t002:** Physical properties of the samples.

Properties	Samples				
	CS0	CS2	CS4	CS6	CS8
Thickness (mm)	0.020	0.021	0.020	0.021	0.020
Opacity	0.916	4.985	5.066	5.700	11.920
Transparency	95.08	79.22	69.81	62.32	56.11
TS (MPa)	75.2 ± 1.6	79.3 ± 2.6	104.7 ± 1.3	101.4 ± 1.8	99.6 ± 1.5
YM (MPa)	1158 ± 36	1607 ± 32	2068 ± 28	1993 ± 32	1957 ± 29
EB (%)	21.8 ± 0.5	16.8 ± 0.6	9.9 ± 0.4	9.2 ± 0.5	8.9 ± 0.3

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
