# Peer review of "Cellulose Nanocrystal Reinforced Chitosan Based UV Barrier Composite Films for Sustainable Packaging"

_polymers, 2020, doi:10.3390/polym12010202_

Round 1

Reviewer 1 Report

The manuscript “Cellulose nanocrystal reinforced chitosan based UV barrier composite films for sustainable packaging” by the authors M. Yadav, K. Behera, Y.-H. Chang and F.-C. Chiu, describes the preparation and characterization of composite films based on cellulose nanocrystal/chitosan (CNC/CS), with potential applications in food packaging industry.

In my opinion, the paper needs some improvements and I recommend the publication in Polymers journal, after major revisions:

2, L.61-63: The affirmation: “Moreover, isolated CNCs do not flocculate with water because of their electrostatic repulsion characteristic on the surface. Thus, CNC suspensions are stable for several months.” isn’t generally true! The flocculation of CNC depends on the type of acid hydrolysis. Please revise the paragraph and add the corresponding references! 2, L. 81: To what “physicochemical properties of CNC/CS”, which aren’t investigated in other manuscripts, do the authors refer? Moreover, please revise “physiochemical properties of CNC/CS” with “physicochemical properties of CNC/CS”! 2, L. 83: What “modified CNC” is used in this study? What do the authors want to say related to “biodegradable/CS-based nanocomposite films”?

p.2, L. 92: Please add information about CS, such as: molecular weight and the deacetylation degree, which are important data for this study!!! Please add some characteristics also for microcrystalline cellulose! In addition, the abbreviation of cellulose microcrystal as CMC, could create confusion with carboxymethyl cellulose (CMC)! Usually, the abbreviation for microcrystalline cellulose is MCC! In my opinion, it would be more correct to use MCC than CMC!  

3, L. 96-110: Please prove the dimensions of CNC (length of 100–500 nm and a diameter of 5–30 nm) and explain how you determined this! 4, L. 124-126: Why the authors choose to conditioned the samples at a humidity of 70 % and a temperature of 25 degree? Did the authors follow an ISO standard? Please add this information! 4, L. 136-138: Is this FESEM or SEM? Please make the correction and add more information regarding the apparatus and method! 4, L. 149. The peaks characteristic to amorphous area appear at 18-19 degrees (2θ), for cellulose! That one from 16.3° which is presented as characteristic to amorphous cellulose, is actually characteristic to crystalline area and corresponding to (10-1) crystallographic plan! 5, L. 181: Why the authors choose the humidity of 75% to conditioned the samples, when the usual ISO standard use an RH of 65%? Please add the ISO standard used by the authors and explain the chosen conditions! 6. L. 201-203. “CMC was totally converted into CNC, which may be confirmed by the presence of two peaks at 1646 and 1241 cm-1 in the CNC spectrum.”??? How did the authors deduce a total transformation of cellulose from FTIR? Please explain all the bands in function of corresponding vibrations! I don’t understand why doesn’t appear in FTIR of chitosan a broad

peak around 3405 cm−1, corresponded to the O–H and N–H stretching vibration, if the samples were conditioned at such high humidity? The section 3.1 should be totally revised and enhanced with more information!

6, L. 213-214: “CMC showed four crystalline peaks at 15.3°, 16.5°, 22.6°, 28.5°, and 34.7°”?? Please revise the sentence and add information regarding each peak and the corresponding crystallographic plane! 6, L. 215-217: How the authors obtained CI of 0.40 and 0.50, when the degree of crystallinity is expressed in percentage, as is presented in Eq. (1)? 6, L. 217-218: In my opinion, there is a problem with the XRD of chitosan, due to the fact that usually, the characteristic peaks appear at 2θ of 10° and 20°, corresponding to the crystallographic planes (002) and (101), and not at 12° and 23° as in this manuscript! In addition, the chitosan is not totally amorphous, so please calculate CI for all samples and demonstrate the modifications which take place by incorporation of CNC! Moreover, the peaks appear in XRD due to crystalline nature, not amorphous nature!! Please revise the sentence! 6-7, L. 220-221: “the peaks of CS and CNC shifted due to changes in the lattice distance”? Please add the information regarding the changes in lattice distance and how these parameters influence the XRD spectra! 7, L. 221: “15.60°, 16.40°, 22.96°, and 34.60°” are not signals, but diffraction peaks! 7, L. 227: There are some ambiguities regarding the morphological characterizationof the samples, through SEM or FESEM! SEM appears in the text, while FESEM appears at Figure 4! What did the authors used? 13. L. 382: Why the authors showed both UV absorbance curve and UV transmittance curve for the same samples?

As a conclusion, there are serious difficulties in the interpretation of data and some of the foundations are unsafe. Overall there are a lot of unclear assumptions and the structure and content of the paper still needs much work!

Author Response

Response to reviewer 1

Comments and Suggestions for Authors

The manuscript “Cellulose nanocrystal reinforced chitosan based UV barrier composite films for sustainable packaging” by the authors M. Yadav, K. Behera, Y.-H. Chang and F.-C. Chiu, describes the preparation and characterization of composite films based on cellulose nanocrystal/chitosan (CNC/CS), with potential applications in food packaging industry.

In my opinion, the paper needs some improvements and I recommend the publication in Polymers journal, after major revisions:

2, L.61-63: The affirmation: “Moreover, isolated CNCs do not flocculate with water because of their electrostatic repulsion characteristic on the surface. Thus, CNC suspensions are stable for several months.” isn’t generally true! The flocculation of CNC depends on the type of acid hydrolysis. Please revise the paragraph and add the corresponding references!

Response: Thanks for valuable suggestion. Author has been modified the sentence accordingly. The incorporated line is given as follows:                                                                                                                     …………Moreover, the flocculation of CNC depends on the type of hydrolysis [27]. Sometime, isolated CNC does not flocculate with water because of their electrostatic repulsion characteristic on the surface. Thus, CNC suspensions are stable for several months.

2, L. 81: To what “physicochemical properties of CNC/CS”, which aren’t investigated in other manuscripts, do the authors refer? Moreover, please revise “physiochemical properties of CNC/CS” with “physicochemical properties of CNC/CS”!

Response: Author studied details about the UV behaviour property of films which is not reported yet by another.                                                                                                                                                                   There are listed many references:                                                                                                                                                       Khan et al., 2012[31], Dehnad et al., 2014[32], Azeredo et al., 2010[33], Wu et la., 2014 [34], Borysiak et al., 2016[35]                                                                                                                                                             The suggested lines were followed accordingly. The modified sentence is provided below:                                                                                                      …………..Among known CS-based composites, CNC/CS nanocomposite is widely investigated [31,32,33,34,35] over the past years. To the best of our knowledge, the UV-barrier properties of CNC/CS bionanocomposites have not been investigated yet.

2, L. 83: What “modified CNC” is used in this study?

Response: Author fabricated CNC from CMC, called modified CNC. Because it contains number of functional which improved its dispersion in water and property of the chitosan composite film.

What do the authors want to say related to “biodegradable/CS-based nanocomposite films”?

Response: As I mentioned in introduction, plastic packaging materials are non-biodegradable in nature are releasing so many hazardous materials which is big issue nowadays for environment. Keeping it in mind author fabricated a green films for packaging which is biodegradable. But sometime our films may come to contact with soil during use. Therefore, the biodegradability behaviour of our film with soil for a for a certain time is necessary to study.

p.2, L. 92: Please add information about CS, such as: molecular weight and the deacetylation degree, which are important data for this study!!! Please add some characteristics also for microcrystalline cellulose! In addition, the abbreviation of cellulose microcrystal as CMC, could create confusion with carboxymethyl cellulose (CMC)! Usually, the abbreviation for microcrystalline cellulose is MCC! In my opinion, it would be more correct to use MCC than CMC! 

Response: Thanks for valuable suggestion. Author has been inserted the suggested information of chitosan and CMC in appropriate part of manuscrict accordingly. Author humbly request to reviewer to accept CMC notation for cellulose microcrystal due to some editing problem in figures. Author will follow reviewer’s suggestion in the next project. The modified part is provided below:

………CS (molecular weight = 350,000 gmol-1 ; the deacetylation degree = 90%) was purchased from TCI, Japan. Cellulose microcrystal (CMC) (size 5m) was supplied by JRS, France. Sulfuric acid and glacial acetic acid were obtained from J.T. Baker, USA. Phosphotungstic acid (staining agent) was used as received from Sigma Aldrich (Taiwan). Deionized water (DIW) was used in the entire study.

3, L. 96-110: Please prove the dimensions of CNC (length of 100–500 nm and a diameter of 5–30 nm) and explain how you determined this!

Response: Author used TEM followed by “Image J software” for calculating the size of CNC. The modified sentence is given below:

……. The product was freeze dried and called CNC. With help of Image J software ( applying on TEM) , the length and diameter of CNC was found 100–500 nm and 5–30 nm, respectively. The obtained CNC yield was ca. 65%.

4, L. 124-126: Why the authors choose to conditioned the samples at a humidity of 70 % and a temperature of 25 degree? Did the authors follow an ISO standard? Please add this information!

Response: Author follow reviewer’s suggestion and rechecked the condition and found that used samples were conditioned at RH by 65% with temperature 25°C. The used ISO standard was valued 7730.

4, L. 136-138: Is this FESEM or SEM? Please make the correction and add more information regarding the apparatus and method!

Response: Author used FESEM for morphology of all samples in whole experiment. The information regarding the apparatus and method was mentioned in below sentence. The modified sentence is given below:

…….The dispersion of CNC in the chitosan matrix and the particle sizes were observed with a Field emission scanning electron microscopy (FESEM) (SEM; Jeol JSM-7500F) operated with an acceleration voltage of 18 kV. All the specimens were sputter-coated with gold before examination.

4, L. 149. The peaks characteristic to amorphous area appear at 18-19 degrees (2θ), for cellulose! That one from 16.3° which is presented as characteristic to amorphous cellulose, is actually characteristic to crystalline area and corresponding to (10-1) crystallographic plan!

Response: Author has been modified the XRD part of manuscript by reviewer’s suggestion accordingly.

5, L. 181: Why the authors choose the humidity of 75% to conditioned the samples, when the usual ISO standard use an RH of 65%? Please add the ISO standard used by the authors and explain the chosen conditions!

Response: Thanks for valuable suggestion. It was missing for RH. Author has been modified RH by 65%. This thermal condition was in the acceptable range of ISO 7730.

L. 201-203. “CMC was totally converted into CNC, which may be confirmed by the presence of two peaks at 1646 and 1241 cm-1 in the CNC spectrum.”??? How did the authors deduce a total transformation of cellulose from FTIR? Please explain all the bands in function of corresponding vibrations! I don’t understand why doesn’t appear in FTIR of chitosan a broad peak around 3405 cm−1, corresponded to the O–H and N–H stretching vibration, if the samples were conditioned at such high humidity? The section 3.1 should be totally revised and enhanced with more information!

Response: The necessary peaks were added for more better explanation for the conversion of CMC to CNC. Author has been modified the interpretation FTIR part accordingly.

The incorporated part is given below:

……….The FTIR spectra of CNC confirmed that a sharp peak at 3350 cm−1 was found due to the OH vibrations in hydrogen bonds [41]. The absorption bands between 2800–3000 cm−1 originated from C–H stretching and bending vibrations [42]. The other absorption bands were assigned at 567 (O-H out of plane bending vibrations), 1058 (C–O stretching at C-3 position), and 1163 cm−1 (C–O-C stretching motion) [43]. Aside from these peaks, the CNCs showed two more peaks at 1645 and 1240 cm−1, confirming the acidification of the CMCs was successfully performed [44]. Furthermore, the abundant oxygen functional groups rendered the CNCs hydrophilic, which improved their solubility in water.

Author supposed the broad peak around 3405 cm−1, corresponded to the O–H and N–H stretching vibration in the chitosan is present. But due to low transmittance (compiling after four sample together) it is hard to identify. Regarding humidity, author modify its condition by 65.

Here, author humbly request to reviewer to accept the presented FTIR as such. But in next project, we will definitely will follow the reviewers suggestions.

6, L. 213-214: “CMC showed four crystalline peaks at 15.3°, 16.5°, 22.6°, 28.5°, and 34.7°”?? Please revise the sentence and add information regarding each peak and the corresponding crystallographic plane!

Response: Thanks for valuable suggestion. Author revised the sentence accordingly. With suggested information, the modified part is given below:                                                                                                                 

The physical interaction between biopolymer and nanofiller can be understood based on degree of crystallinity. XRD analysis was conducted within the 2ϴ range of 10° to 40° on CMC, CNC, CS0, CS2, CS4, CS6, and CS8 samples. As shown in Figure 3, CMC showed four crystalline peaks at 15.3°, 16.5°, 22.6° and 34.7°. However, after acid treatment of CMC, CNC exhibited peaks at 15.6° (101), 16.4° (10-1), 23.0° (200), and 34.6° (400) [7,45]. The calculated crystallinity index (CI) values of CMC and CNC were ca. 40% and 50%, respectively, suggesting that structural change by the acid treatment of CMC was responsible for the increased CI of CNC. For CS, two main peaks appeared at 10°(002) and 20°(101) due to its crystalline nature. The CI of chitosan film was found 35% which was agreed with previously published paper [46]. With the addition of CNC to the CS matrix, the mechanical and physical properties of the films were changed due to the crystalline morphology, crystal forms, and crystallite size of CNC [47]. In the case of blends the diffraction peaks for both chitosan and CNC are sifted to higher value with increasing the CNC content, thus the peaks from 10° is shifted to 12.05°, 15.6° at 15.7°, 16.4° at 16.7°, 20° is combined with that from 23° of CNC and is shifted at 22.75° and the signal at 34.6° at 34.65°. These shifting induce an increasing in the lattice distance with increasing the CNC content. The incorporation of the rigid molecules of CNC in the chitosan matrix induces an orientation of the polymeric molecules and favour the formation of hydrogen bonds between the polymeric chains.

6, L. 215-217: How the authors obtained CI of 0.40 and 0.50, when the degree of crystallinity is expressed in percentage, as is presented in Eq. (1)?

Response: Thanks for valuable suggestion. The CI of CMC and CNC was found 40% and 50%, respectively, by using eq. (1). Author has been modified the presented sentence:

……….The calculated crystallinity index (CI) values of CMC and CNC were ca. 40% and 50%, respectively, suggesting that structural change by the acid treatment of CMC was responsible for the increased CI of CNC.

6, L. 217-218: In my opinion, there is a problem with the XRD of chitosan, due to the fact that usually, the characteristic peaks appear at 2θ of 10° and 20°, corresponding to the crystallographic planes (002) and (101), and not at 12° and 23° as in this manuscript! In addition, the chitosan is not totally amorphous, so please calculate CI for all samples and demonstrate the modifications which take place by incorporation of CNC! Moreover, the peaks appear in XRD due to crystalline nature, not amorphous nature!! Please revise the sentence!

Response: Thanks for valuable suggestion. Author modified the XRD pattern of chitosan according to reviewer’s suggestion. The characteristic peaks of chitosan appeared at 2θ of 10°(002) and 20°(101) due to its crystalline nature.The CI of chitosan film was found 35% which is agreed with previously published paper [46]. The other samples CS0.CS2,CS4,CS6 and CS8 showed the CI at 37%,38%, 40% and 42% (results not shown in manuscript).

[46] Sakurai, K.,Takagi, M.,Takahashi, T.Crystal structure of chitosan. I. Unit cell parameters. SEN-l GAKKAISHI, T-246-53. Vol. 40, page 114-121,No. 7 (1984)

The modified part of XRD is appended below:

3.2. XRD

…………The physical interaction between biopolymer and nanofiller can be understood based on degree of crystallinity. XRD analysis was conducted within the 2ϴ range of 10° to 40° on CMC, CNC, CS0, CS2, CS4, CS6, and CS8 samples. As shown in Figure 3, CMC showed four crystalline peaks at 15.3°, 16.5°, 22.6° and 34.7°. However, after acid treatment of CMC, CNC exhibited peaks at 15.6° (101), 16.4° (10-1), 23.0° (200), and 34.6° (400) [7,45]. The calculated crystallinity index (CI) values of CMC and CNC were ca. 40% and 50%, respectively, suggesting that structural change by the acid treatment of CMC was responsible for the increased CI of CNC. For CS, two main peaks appeared at 10°(002) and 20°(101) due to its crystalline nature. The CI of chitosan film was found 35% which was agreed with previously published paper [46]. With the addition of CNC to the CS matrix, the mechanical and physical properties of the films were changed due to the crystalline morphology, crystal forms, and crystallite size of CNC [47]. In the case of blends the diffraction peaks for both chitosan and CNC are sifted to higher value with increasing the CNC content, thus the peaks from 10° is shifted to 12.05°, 15.6° at 15.7°, 16.4° at 16.7°, 20° is combined with that from 23° of CNC and is shifted at 22.75° and the signal at 34.6° at 34.65°. These shifting induce an increasing in the lattice distance with increasing the CNC content. The incorporation of the rigid molecules of CNC in the chitosan matrix induces an orientation of the polymeric molecules and favour the formation of hydrogen bonds between the polymeric chains.

6-7, L. 220-221: “the peaks of CS and CNC shifted due to changes in the lattice distance”? Please add the information regarding the changes in lattice distance and how these parameters influence the XRD spectra!

Response: Author modified the manuscript according to reviewer changes.

3.2. XRD………

…………In the case of blends the diffraction peaks for both chitosan and CNC are sifted to higher value with increasing the CNC content, thus the peaks from 10° is shifted to 12.05°, 15.6° at 15.7°, 16.4° at 16.7°, 20° is combined with that from 23° of CNC and is shifted at 22.75° and the signal at 34.6° at 34.65°. These shifting induce an increasing in the lattice distance with increasing the CNC content. The incorporation of the rigid molecules of CNC in the chitosan matrix induces an orientation of the polymeric molecules and favour the formation of hydrogen bonds between the polymeric chains.

7, L. 221: “15.60°, 16.40°, 22.96°, and 34.60°” are not signals, but diffraction peaks!

Response: The corrections were performed accordingly in appropriate place of manuscript.

7, L. 227: There are some ambiguities regarding the morphological characterizationof the samples, through SEM or FESEM! SEM appears in the text, while FESEM appears at Figure 4! What did the authors used?

Response: Author used FESEM for morphology in whole experiment. For consistency, author used FESEM word in the whole manuscript accordingly.

L. 382: Why the authors showed both UV absorbance curve and UV transmittance curve for the same samples?

Response: The UV absorbance curve used for studying the absorption of UV light by chitosan and composite films. To avoid the oxidative rancidity catalysed by UV light, the packaging film could able to absorb UV region.                                                                                                                                                

On the other hand, the transmittance curve of used films confirmed the transparency and blocking effect. It can be correlated to the CNC dispersion in chitosan matrix. The films showed higher transmittance values at the range of visible light (400–800 nm) than that at the range of ultraviolet light (200–400 nm). So, author incorporated both curve for studying details about UV.         

Recently, a published paper by Balakrishnan et al. studied the similar results. Please, follow the provided link for details: DOI 10.1002/star.201700139

As a conclusion, there are serious difficulties in the interpretation of data and some of the foundations are unsafe. Overall there are a lot of unclear assumptions and the structure and content of the paper still needs much work!

Response: Thanks for valuable suggestions. Author has been modified the conclusion according to reviewer suggestions and try to avoid the unclear assumption/structure/content of the work. Author is still studying some more information regarding assumptions, structure and content for the next project. Therefore, author humbly request to reviewer to consider manuscript with this provided modified conclusion as such.

The Modified conclusion is provided below:

CNC-reinforced CS-based sustainable biocomposite films with enhanced properties were successfully obtained by mixing CNC and CS through solution casting. The needle-shaped CNC was successfully synthesized from CMC by acid-hydrolysis treatment. The modification of the chemical structure of CNC/CS composite was analyzed by FTIR and XRD studies, revealing the formation of hydrogen bonds between CNC and CS. FESEM and OM images confirmed a dispersion of CNC throughout the CS matrix without remarkable agglomerations. TGA results indicated that chitosan and its composites exhibited almost same thermal stability. In the presence of CNC in the nanocomposites, water and UV barrier properties were also improved. FWS and EMC were reduced by 27% and 32%, respectively, after incorporation of only 4 wt.% CNC. The aim of improving UV absorption with the presence of CNC in composites was achieved, and the composites evidently showed lower transmittance compared with control CS. The TS and YM of CS film was improved with the incorporation of CNC, whereas EB decreased with increasing brittleness of the composite films. Moreover, the composites with CNC loading showed lower biodegradation percentage than the control CS due to the presence of strong interaction through H-bonding between CS and CNC. Therefore, CNC/CS bionanocomposite film could be candidates for potential applications in the food packaging field.

Reviewer 2 Report

My Comments :

Yadav et al., fabricated chitosan based composite films via casting blend technique. FTIR, XRD, SEM, and TEM confirmed the fabrication of composite films. The addition of only 4 wt.% CNC in the CS film improved the tensile strength and Young’s modulus by up to 39% and 78%, respectively. Depending on CNC content, the moisture absorption decreased by 34.1%–24.2% and the water solubility decreased by 35.7%–26.5% for the composite films compared with neat CS film. The water vapor permeation decreased from 3.83 ×10 -11 gm -1 s -1 Pa -1 to 2.41×10 -11 gm -1 s -1 Pa -1 in the CS-based films loaded with (0–8 wt.%) CNC. The water and UV barrier properties of the composite films showed better performance than those of neat CS film. Authors have well interpreted their research work. But, still few part of the Minor revision is needed before it is accepted for  publication.

The method of sample preparation for TEM is required. The formula for tensile strength/E.B. should be inserted or kindly cite some papers related to it. Line 474-476, Kindly state the reason behind biodegradability this will aid in for better understanding about the film property. The stability of chitosan film in water should be referred with recent papers. Kindly reedit the Conclusion part with more technical results.

Author Response

Response to reviewer 2

Comments and Suggestions for Authors

My Comments:

Yadav et al., fabricated chitosan based composite films via casting blend technique. FTIR, XRD, SEM, and TEM confirmed the fabrication of composite films. The addition of only 4 wt.% CNC in the CS film improved the tensile strength and Young’s modulus by up to 39% and 78%, respectively. Depending on CNC content, the moisture absorption decreased by 34.1%–24.2% and the water solubility decreased by 35.7%–26.5% for the composite films compared with neat CS film. The water vapor permeation decreased from 3.83 ×10 -11 gm -1 s -1 Pa -1 to 2.41×10 -11 gm -1 s -1 Pa -1 in the CS-based films loaded with (0–8 wt.%) CNC. The water and UV barrier properties of the composite films showed better performance than those of neat CS film. Authors have well interpreted their research work. But, still few part of the Minor revision is needed before it is accepted for  publication.

The method of sample preparation for TEM is required. The formula for tensile strength/E.B. should be inserted or kindly cite some papers related to it. Line 474-476, Kindly state the reason behind biodegradability this will aid in for better understanding about the film property. The stability of chitosan film in water should be referred with recent papers. Kindly reedit the Conclusion part with more technical results.

Response: Thanks for valuable suggestion. The method of sample preparation for TEM is added in manuscript at appropriate place with proper references. The formula for tensile strength/E.B is cited with current published papers. In line 474-476, author inserted proper references for the reason behind biodegradability property of the film. The stability of chitosan film in water is referred with recent papers. The conclusion part is improved with more technical results.

The modified conclusion is appended below:

CNC-reinforced CS-based sustainable biocomposite films with enhanced properties were successfully obtained by mixing CNC and CS through solution casting. The needle-shaped CNC was successfully synthesized from CMC by acid-hydrolysis treatment. The modification of the chemical structure of CNC/CS composite was analyzed by FTIR and XRD studies, revealing the formation of hydrogen bonds between CNC and CS. FESEM and OM images confirmed a dispersion of CNC throughout the CS matrix without remarkable agglomerations. TGA results indicated that chitosan and its composites exhibited almost same thermal stability. In the presence of CNC in the nanocomposites, water and UV barrier properties were also improved. FWS and EMC were reduced by 27% and 32%, respectively, after incorporation of only 4 wt.% CNC. The aim of improving UV absorption with the presence of CNC in composites was achieved, and the composites evidently showed lower transmittance compared with control CS. The TS and YM of CS film was improved with the incorporation of CNC, whereas EB decreased with increasing brittleness of the composite films. Moreover, the composites with CNC loading showed lower biodegradation percentage than the control CS due to the presence of strong interaction through H-bonding between CS and CNC. Therefore, CNC/CS bionanocomposite film could be candidates for potential applications in the food packaging field.

Round 2

Reviewer 1 Report

The authors paid attention to all comments and the new and improved version of the manuscript can be published in Polymers journal.